# Effects of Typical Solvents on the Structural Integrity and Properties of Activated Kaolinite by Wet Ball Milling

**DOI:** 10.3390/nano12234255

**Published:** 2022-11-29

**Authors:** Shunjie Luo, Yang Chen, Weiting Xu, Jiangxiong Wei, Zhaoheng Li, Shiqing Huang, Haoliang Huang, Junlu Zhang, Qijun Yu

**Affiliations:** 1School of Materials Science and Engineering, South China University of Technology, Guangzhou 510640, China; 2Guangdong Research Institute of Water Resources and Hydropower, Guangzhou 510635, China; 3Guangdong Provincial Academy of Building Research Group Co., Ltd., Guangzhou 510500, China; 4School of Mechanics and Construction Engineering, Jinan University, Guangzhou 510632, China

**Keywords:** kaolinite, organic solvents, activated, wet ball milling, intercalation, buffering effect

## Abstract

The influence of organic solvents on the structural integrity and properties of activated kaolinite were compared and analyzed via characterization techniques and molecular dynamics (MD) simulation. The results revealed that the organic intercalators can be easily inserted into the interlayer spaces of activated kaolinite within a short time of the wet ball milling. The DMSO intercalated kaolinites maintained structural integrity due to the high intercalation rate and the excellent buffering effect against the crushing force of milling during the delamination/exfoliation process. The delaminated layers of the DMSO–kaolinite complex exhibited a high specific surface area of 99.12 m^2^/g and a low average thickness of 35.21 nm. The calculated elastic properties of the organo-kaolinite complex manifested the intercalation of DMSO into a kaolinite interlayer, which could improve the compressibility and structural integrity of kaolinite nanosheets. The DMSO–kaolinite complex was easier to peel off when compared to the other organic intercalators due to its more intercalated molecules.

## 1. Introduction

Kaolin is a fundamental clay mineral widely applied in the industrial production fields of sewage treatment [1], paper making, ceramics, etc. Kaolin often functions as an adsorbent, catalyst, degradation agent and filler for the modification of rubber and polymers [2,3,4,5]. The primary component in kaolin is kaolinite with a unit structure of 1:1 layered aluminosilicate, which is composed of AlO_2_OH_4_ octahedron and SiO_4_ tetrahedron [6]. The aluminosilicate layers are bonded with hydrogen bonding force [7]. The original kaolinite is barely able to form a stable dispersion in solvents. As such, it cannot be qualified for the purposes of enhancing the physicochemical performance of a composite material, due to its low specific surface area, thick lamellae and uneven dispersion in solvents, which is caused by the strong cohesive energy between its layers.

It has been widely accepted that delamination/exfoliation is a vital industrial treatment for the purposes of enlarging the layer spacing of aluminosilicate layers, reducing the particle size, increasing the surface area and thus improving the applicability of kaolinite. Extensive studies have been conducted on inserting small organic molecules between the layers of kaolinite in order to expand the interlayer spacing to obtain nanocomposites with separation properties [8,9,10]. The commonly adopted intercalating methods include high pressure processing, melting and shaking—which rely heavily on the temperature, pressure and equipment conditions [11,12,13,14]. Moreover, disadvantages to these approaches are apparent in the long-term aging and mixing processes before the peeling, the approaches were hardly applied in actual production for long cycles and high energy consumption [15].

The intercalation coupled with a mechanical grinding and stirring treatment was proven to be an effective approach for achieving delamination/exfoliation. Recent studies have reported that the intercalation treatment of kaolinite with certain sorts of intercalation molecules can break the interlayer hydrogen bonds. This, therefore, increases the interlayer spacing of the layered kaolinite particles [8]. Certain alkali salts, organic molecules or polymers can be inserted between the kaolinite layers by forming hydrogen bonds with the oxygen atoms of the silicon–oxygen surface or the hydroxyl groups of the aluminum oxide layer in order to delaminate the layered structure [9]. The properties regarding specific surface area, adsorption and dispersibility of kaolinite can be fundamentally improved in order to endow kaolinite with new functions and unique properties. Meanwhile, during the intercalation process, mechanical grinding and stirring accelerate the breakage of strong cohesion bonding between the kaolinite layers, thereby decreasing the kaolinite particle size to a micron scale, which is achieved without damaging the crystalline structure and lamellar morphology. In addition, the ground kaolinite particles expose more intercalating sites in order to form new hydrogen bonds with intercalating molecules, thus facilitating the disintegration of the layered stacking structure of kaolinite.

Recent studies have successfully synthesized the organo-kaolinite complexes using acidified kaolinite and organic intercalator solvent via the assistance of stirring and wet ball milling [16,17,18,19]. However, the effects of the typical organic intercalators on the structural integrity and properties of kaolinite with wet ball milling, as well as the reasons for the performance variations of these organo-kaolinite complexes, have rarely been reported. In this paper, kaolinite was firstly subjected to an acid activation process and then immersion into three commonly available organic solvents, i.e., potassium acetate (KAc), formamide (FA) and dimethyl sulfoxide (DMSO) for the purposes of intercalating. After that, the suspensions of organo-kaolinite complexes were grounded via wet ball milling. The structural morphology and layer spacing of the synthesized organo-kaolinite complexes were investigated and compared by XRD, FTIR, SEM, TGA and nitrogen adsorption. The basis for the property variation of the synthesized complexes was revealed by molecular dynamics simulation.

## 2. Materials and Methods

### 2.1. Materials

The kaolinite sample used in the study was commercially available from Maoming City, Guangdong Province, China. The chemical composition is tabulated in Table 1. The DMSO, FA and absolute alcohol were purchased from Tianjin Fuyu Fine Chemical Company. The KAc was obtained from Richjoint company, Shanghai, China. The hydrochloric acid with a concentration of 36~38% was supplied from Guangshi reagent technology company, Guangdong Province, China. The zircon balls, used as the grinding media during the milling process, composed of zirconium dioxide ca. 95% and antimony trioxide ca. 5% by mass, were acquired from Yike Abrasive Materials Company, Shenzhen, China.

### 2.2. Intercalation Process

In order to prepare the acidified kaolinite, 20 g of raw kaolin was immersed in 200 mL of hydrochloric acid solution with a concentration of 5 mol/L. The kaolinite suspension was continuously stirred for 6 h and then rinsed with water until neutral. The suspension was dried at 80 °C for 24 h in order to gain the acidified kaolinite sample. There were three sorts of saturated solvents, i.e., KAc, FA and DMSO solvents, which were prepared as the intercalation agents. As for the intercalation process itself: A total of 10 g of acidified kaolinite was mixed with the saturated KAc, FA and DMSO solvents in order to form organo-kaolinite complex solutions; then, the solutions were subjected to wet ball milling in order to facilitate the intercalation process. The intercalation and exfoliation process are illustrated in Figure 1. The stirred ball mill (WTO-3) was used for grinding, as well as for enhancing the collision, friction and shearing interactions of the kaolinite particles [20]. The zircon balls, with a diameter range from 0.8 to 1.0 mm, were used as the grinding media. All the solvents were agitated and stirred for 2 h at the speed of 200 rpm in the ball mill. After the wet ball milling, the zircon balls and the ground slurry were separated via sieving. Further, the delaminated kaolinite was recovered by filtration from the ground slurry. The organo-kaolinite complex solutions were rinsed with absolute alcohol and dried in a vacuum at 80 °C for 24 h in order to gain the final products. The obtained various intercalated kaolinites denoted KAc–kaolinite, FA–kaolinite and DMSO–kaolinite, respectively.

### 2.3. Materials Characterization

The chemical composition of the raw kaolinite and the HCl-treated kaolinite samples were determined by X-ray fluorescence spectrometry (XRF) on the wavelength dispersive spectrometer (Axios, PW4400 PANalytical, 2015, Heracles Almelo, The Netherlands). For the purposes of analyzing the interlayer spacing and intercalation rate, the organo-kaolinite complex samples were subjected to powder X-ray diffraction (X’pert Powder, PANalytical). This was conducted with Cu-Kα radiation at the step size of 0.02° and count time of 0.2 s in the scanning range of 2*θ* between 5° and 90°, running at 40 kV and 40 mA. The change in the chemical properties of kaolinite before or after stripping was characterized by Fourier transform infrared spectroscopy (VECTOR-22IR, Bruker Instruments Corp., Billerica, MA, USA). The powder samples of the tests were dried and mixed with KBr for tablet processing. The conditions of the infrared test ranged from 400 cm^−1^ to 4000 cm^−1^, the test resolution was 4cm^−1^ and each test was scanned 64 times [21]. Scanning electron microscopy (SEM) morphologies of the gold-coated powder samples were performed using the device of NOVA NANO SEM 430 with a field emission gun, a magnification of 30× up to 300,000 and the resolution for Gun. 0.004–20 nA, FEI Company, Netherlands. The kaolinite samples were dispersed in ethanol with ultrasound for the purposes of transmission electron microscopy (TEM) analysis. Then, its dispersion was dropped on carbon-coated copper TEM grids with a mesh size of 40 μm × 40 μm. Further, the samples were dried under vacuum before test. TEM measurements were performed on a JEOL model 1400EX electron microscope operated at an accelerating voltage of 120 kV. The thermogravimetric analysis (TGA, Netzsch STA449F5, Gerätebau, Germany) was performed from 30 to 1000 °C at a heating rate of 10 °C/min^−1^. Nano particle sizes were tested by their laser particle size and then zeta potential was analyzed (Nano ZS90, Malrern Instruments Corp., Britain). The specific surface area (SSA) values were obtained with an automatic system (Model No. 2200 A, Micromeritics Instrument Corp., Norcross, GA, USA) in a liquid-nitrogen atmosphere, using the BET method. Before measurement, the samples were pre-heated at 130 °C under nitrogen for 24 h. The SSA was calculated by the BET equation, and the total pore volumes were evaluated from a nitrogen uptake at a relative pressure of ca. 0.99. In order to measure the size and thickness of obtained kaolinite nanosheets, an atomic force microscope (Bruker MultiMode 8) with a Si_3_N_4_ probe (radius = 30 μm) was adopted. The preparation of the measurement samples was performed by dropping diluted kaolinite suspensions (0.1 g/L) on the mica sheet. Subsequently, the mica sheet was rotated with 500 rpm for 10 min and then dried at room temperature in order to avoid the coagulation and overlapping of particles. In order to obtain an accurate thickness and size distribution of different kaolinite samples, 20 sheets of each sample were subjected to the AFM measurement, which was equipped with NanoScope Analysis 1.8 software for the purposes of statistical analysis.

### 2.4. Models and Simulation Details

A kaolin unit cell with the chemical composition of A1_4_Si_4_O_10_(OH)_8_ [22] was constructed as the basic structural unit for the simulation. The lattice parameters of the unit cell with P1 symmetry were a = 5.154 Å, b = 8.942 Å, c = 7.391 Å, α = 91.93°, β = 105.05° and γ = 89.80°. The periodic supercell containing 16 (4 × 4 × 1) unit cells was established [23]. The intercalation molecules (KAc, FA and DMSO) were separately packed into three boxes with the lengths a and b exactly matching the size of the established kaolinite supercell. The amounts of the intercalation molecules for packing in the model were calculated according to the TGA testing results. The sandwich complex structure models with the upper and lower layers of kaolinite, as well as the middle layer of intercalation molecules, were built for the following simulation work.

The molecular dynamics simulations were conducted employing the Forcite program and Compass II force field from the Material Studio software. The geometry optimization was performed for all the complex structure models prior to the dynamic calculations. The dynamic calculations were run in the NPT ensemble for 300 ps with the fixed pressure of 0.1 Mpa and the fixed temperature of 298 K [24,25]. The pressure and temperature were regulated by the Nose thermostat and Berendsen barostat, respectively. Following NPT simulation, a further NVT ensemble with a simulation time of 300 ps was performed. The time step of the simulation was set 1.0 fs and the trajectory frame was recorded every 20 fs. The electrostatic interaction and the van der Waals of the particles was calculated by the Ewald and atom-based summation method, respectively. During the simulations, all the atoms in the complex system moved freely in three dimensions.

## 3. Results

### 3.1. Mineralogy Analysis

The XRD patterns of the original kaolinite, i.e., in regard to the acidified kaolinite and the organo-kaolinite complexes, are plotted in Figure 2. It can be seen that there are two characteristic peaks of kaolinite at 2*θ* = 12.5° and 27.6°, thereby representing two typical reflections at the (001) and (002) crystal planes, which corresponds to d-spacings of 7.167 Å and 3.573 Å, respectively. It is apparent that kaolinite is the only clay mineral in the original soil [26]. The acidified kaolinite presents a higher diffraction peak intensity and a narrower *d*-spacing of 7.164 Å than that of the original kaolinite [27]. This may be attributed to the increase in hydrogen bonds on the surface of kaolinite after the acidification treatment. Combined with the XRF testing results, as shown in Table 2, the original kaolinite and the acidified kaolinite exhibit visible differences in SiO_2_ and Al_2_O_3_ content. The reduction in Al_2_O_3_ content of acidified kaolinite demonstrates that the acidification treatment rinse off the impurity oxides in kaolinite [28]. As seen in Figure 2, the XRD pattern of the KAc–kaolinite complex via wet ball milling present lower peak intensities when compared to the original kaolinite but a new characteristic peak barely appears. However, in regard to the FA–kaolinite and DMSO–kaolinite complex that were subjected to wet ball milling, characteristic peaks appear at 2*θ* = 5° to 10°, thereby indicating DMSO and FA can be easily intercalated into the intralayer of the acidified kaolinite. The d-spacing of the DMSO–kaolinite complex is 11.233 Å and larger than that of FA–kaolinite (10.225 Å). This is attributed to the fact that the total inserted molecule volume of DMSO is larger than that of FA; hence, leading to a greater interlayer spacing between kaolinite sheets [29,30,31].

The intercalation efficiency (*E_i_*) is proposed in order to compare the *d_001_* intensities of the original kaolinite spacing (*I_KOS_*) with that of the intercalated kaolinite spacing (*I_KC_*), as per the Equation (1) [21]:(1)$$E_{i}=\frac{I_{kc}}{I_{KOS}+I_{KC}}\times 100\%$$ As seen in Figure 2, it is apparent that the peak intensity corresponding to the (001) crystal plane of the DMSO–kaolinite complex is higher than that of the FA–kaolinite complex. Further, the intensity of the characteristic peak at the (002) crystal plane of the DMSO–kaolinite complex is lower when compared to that of the FA–kaolinite complex [30]. From viewing the calculation results, the intercalation efficiency of the DMSO–kaolinite complex is 72.8% and, thus, higher than that of the FA–kaolinite complex (40.6%). The results demonstrate that DMSO is preferable to FA when being inserted into a kaolinite layer.

### 3.2. FTIR Analysis

The FTIR spectra of original kaolinite, acidified kaolinite and organo-kaolinite complexes by wet ball milling are depicted in Figure 3. The characteristic peak at 1084 cm^−1^ and 542 cm^−1^ refers to the stretching vibration and the bending vibration of silicon–oxygen bonds, respectively. The characteristic peaks appearing at 471 cm^−1^, 3697 cm^−1^, 3672 cm^−1^ and 3652 cm^−1^ are attributed to the same direction and the two different vibrations of the interlayer hydrogen bonding Al-OH. The characteristic peak at 3622 cm^−1^ relates to the structural unit layer tetrahedral sheet and the octahedral sheet junction in kaolinite_,_ i.e., the tensile vibration bonging between oxygen and hydrogen on the intralayer surface. It is noteworthy that the tensile vibration characteristics of the hydrogen-bonded hydroxyl groups between the kaolinite layers are affected by the inserted guest molecule [31]. As seen in Figure 3, the Al-OH group inside the kaolinite structural unit at 3622 cm^−1^ cannot be affected by acidification treatment. After acidizing, the peak intensities at 3672 cm^−1^ and 3652 cm^−1^ are reduced, due to the precipitation of alumina, which breaks the hydrogen bonding between the layers within structural unit and is conducive to providing more sites for insertion. In addition, the 3663 cm^−1^ result in Figure 3 may be due to the large amount of intercalated DMSO, which render the original hydrogen bonds broken and thus new hydrogen bonds are formed. The characteristic peak appearing at 946 cm^−1^ corresponding to the flexural vibration is attributed to the kaolinite structure change caused by acidification. The precipitation of alumina increases the relative content of silica and is the silicon–oxygen bond stretching at the vibration peak and the flexural vibration peak after acidification [10,32,33]. The intercalators affect the bending vibration of the Al-OH group at 900 cm^−1^ to 950 cm^−1^. The intensity of the band around 946 cm^−1^ and 911 cm^−1^ were significantly weakened due to the inserted intercalators in the intralayer spacing. The organo-kaolinite complex with the higher intercalation rate can effectively alleviate the crystal damage caused by wet ball milling. It is worth noting that the peak intensity of the KAc–kaolinite complex at each position is weaker than those of the other two intercalated kaolinite, thereby indicating that the unit structure of the KAc–kaolinite complex is more inclined to be damaged during the milling process. The DMSO molecule has higher solubility, higher polarities and larger molecular mass than FA. Therefore, DMSO is more inclined to be inserted between the kaolinite layers [34,35]. The DMSO intercalated kaolinite exhibited excellent shear force resistance, thereby weakening the hydrogen bonding force between the layers and mitigating the crushing damage of the kaolinite surface that was induced by milling.

### 3.3. Thermal Analysis

The thermal stability of the organo-kaolinite complexes after wet ball milling were investigated via TG/DTA analysis; the testing results of which are shown in Figure 4. The thermal stability of kaolinite after wet ball milling in DMSO is better than formamide and potassium acetate saturated. The main endothermic peak around 488 °C in the DTG thermal analysis diagram is due to the dehydroxylation of the crystal lattice, which leads to the formation of meta-kaolinite with the loss of water in the layer. The organic molecules residue on the kaolinite surface can be rinsed off with an alcohol washing process after the intercalation process. Therefore, the weight loss of organic molecules calculated by the TG/DTA testing results represents the content of the inserted organic molecules between the kaolin layers. The organic molecule weight loss of KAc, FA and DMSO were 1.5%, 5.74% and 11.22%, respectively. It is apparent that compared with KAc and FA, DMSO presents the best intercalation performance under the condition of acidification and wet ball milling.

### 3.4. Morphology Analysis

The kaolinite surface morphology, particle size and thickness before and after the intercalation process are determined by SEM and TEM testing. The SEM and TEM morphologies of original kaolinite, acidified kaolinite and organo-kaolinite complexes by wet ball milling are shown in Figure 5 and Figure 6, respectively. Figure 5a clearly shows that the original kaolinite had multi-layer stacked irregular polygonal plate structure with a certain thickness. Compared with the original kaolinite, the particle size and the thickness of the KAc–kaolinite complex after wet ball milling (Figure 5b) are reduced. Furthermore, many crushed small plate-like fragments attached on the surface, which lead to a rougher surface. Many cracks appear on the KAc–kaolinite complex lamella surface. This was caused by wet ball milling damage, as shown in Figure 6b. During the wet ball milling process, the shearing force by the grinding media becomes the lateral force on the kaolinite lamella surface, thereby imposing a continuous destruction on the layers and resulting in crushed fragments become attached to the surface. As for the FA–kaolinite complex (as shown in Figure 5c), the kaolinite flake surface was relatively smooth after wet ball milling and retained some small flake fragments without being peeled off. It is seen in Figure 6c that the thickness of the FA–kaolinite complex becomes thinner than that of the original kaolinite. Moreover, small cracks appear on the surface of the partially peeled flake. In the case of the DMSO–kaolinite complex, as seen in Figure 5d, most of the kaolinite lamella surfaces after wet ball milling are the smoothest among all the tested kaolinite variants. The DMSO–kaolinite complex presents a more apparent lamellar structure, thinner thickness and higher structural integrity than the previous two organo-kaolinites. Only a few micro cracks were observed in Figure 6d, demonstrating that the insertion of DMSO molecules between the kaolinite layers can buffer the crushing force and effectively alleviate the structural damage caused by wet ball milling. The results reveal that DMSO has a higher polarity and larger molecule volume. In addition, it exhibited a higher intercalating performance; hence, leading to better buffer and extrusion effects than the other two organic intercalators.

### 3.5. Nano Particle Properties

The average particle size of the original kaolinite, the KAc–kaolinite complex, the FA–kaolinite complex and the DMSO–kaolinite complex by wet ball milling was 3528 nm, 556.1 nm, 836.21 nm and 1171.92 nm, respectively. The average particle size of the DMSO–kaolinite complex with a high intercalation degree is larger than that of the FA–kaolinite. It is evident that the intercalation treatment aggravates grindability of the lamellar-structural kaolinite; further, the inserted intercalator was observed to possess a high content of molecules buffer milling damage on the kaolinite sheets. The average thickness values of the kaolinite samples were calculated by AFM, as shown in Figure 7. The thickness of the original kaolinite soil, the KAc–kaolinite complex, the FA–kaolinite complex and the DMSO–kaolinite complex by wet ball milling were 138.02 nm, 80.91 nm, 53.83 nm and 35.21 nm, respectively. The original kaolinite is a multi-layer stacked structure with a color gradient in thickness. In regard to the KAc–kaolinite complex, the milling damage from the outer layer to the inner layer results in a decrease in thickness. Due to the high peeling and intercalation degree of the DMSO, the thickness of the DMSO–kaolinite complex lamella is sharply reduced and the integrity of the peeled lamella is higher—which, as a results, are also in line with the previous testing results. The BET surface area of the original kaolinite soil, the KAc–kaolinite complex, the FA–kaolinite and the DMSO–kaolinite complex by wet ball milling were 14.03 m^2^/g, 37.07 m^2^/g, 69.88 m^2^/g and 99.12 m^2^/g, respectively. The organic polymer intercalation combined with the wet ball milling process rendered the kaolinite lamella to peel off and for the particle size to reduce. Therefore, the intercalated kaolinite has a higher specific surface area and thinner thickness when compared to the original kaolinite. Among the organo-kaolinite complexes, the DMSO–kaolinite complex exhibits the thinnest thickness and the highest specific surface area due to the high intercalation rate of DMSO.

### 3.6. Molecular Dynamics Simulation

#### 3.6.1. Radial Distribution Function (RDF)

The RDF for particles A to B refers to the distribution density probability of particles B around A, at a given distance. The first sharp peak of the RDF diagram represents the first coordination layer position of the target atoms to be analyzed around the center atoms. The RDF for hydroxyl hydrogen atoms of the alumina octahedral surface of kaolinite to the oxygen atoms of KAc, FA and DMSO are shown in Figure 8a. The RDF calculated for the oxygen atoms of the silicon tetrahedral surface of kaolinite to the hydrogen atoms of KAc, FA and DMSO are shown in Figure 8b.

As seen from Figure 8a, the hydroxyl hydrogen atoms of the kaolinite Al-OH surface have a close coordination with the oxygen atoms of organic intercalators in order to form strong hydrogen bonds. The coordination position of the oxygen atoms of KAc, FA and DMSO was statistically located at 1.37 Å, 1.55 Å and 1.71 Å, respectively. The results reflected that the strength of interaction between the hydroxyl hydrogen atoms of kaolinite follows the order of KAc > FA > DMSO. The intercalated KAc were more closely coordinated with the kaolinite interlayer, while the insertion molecules of KAc were less than those of FA and DMSO, thereby implying KAc intercalated kaolinite lamella is more difficult to peel off. On the contrary, more intercalated DMSO molecules are coordinated with the interlayer of kaolinite, while the coordination position of oxygen atoms is farther than that of KAc and FA. This arrangement of DMSO oxygen atoms manifests that the exfoliation of the DMSO–kaolinite complex is more easily compared to the KAc and FA intercalated kaolinite complexes.

In respect to Figure 8b, it can be seen that the first sharp peak of the RDF diagram for the KAc–kaolinite complex, FA–kaolinite complex and DMSO–kaolinite complex appears at 2.75 Å, 2.79 Å and 2.49 Å, respectively. This suggests that the hydrogen atoms of DMSO form slightly stronger hydrogen bonds with the oxygen atoms of the silicon tetrahedral surface of kaolinite when compared to those of FA and KAc. In regard to all of the intercalated kaolinite complexes, they also presented relatively low broad peaks at some distance from the first peak, implying some of the hydrogen atoms of the organic intercalator were weakly coordinated with the Si-OH surface of kaolinite. The above RDF results are in line with the FTIR analysis results.

#### 3.6.2. Mean Square Displacement (MSD)

The MSD is a measure of the deviation of the particle’s position relative to the reference position after moving over the simulation time. It is dynamic information for the mobility of the interested atom. The MSDs for the oxygen atoms and the hydrogen atoms of KAc, FA and DMSO are plotted in Figure 9, respectively. As for the MSD results of the oxygen atoms that are shown in Figure 9a,c,e, in the x, y and z dimension, the oxygen atoms of KAc and FA retain a relative stable and low MSD value, while the oxygen atoms of DMSO keep mobile in the three dimensions and the MSD value is higher than that of KAc and FA in the z dimension. It is apparent that the oxygen atoms of KAc and FA form more stable hydrogen bonding with hydroxyl hydrogen atoms of the Al-OH surface of kaolinite compared to that of DMSO. When comparing the mobility of hydrogen atoms of organic intercalators, as seen in Figure 9b,d,f, the MSD value of the hydrogen atoms of the intercalators in the z dimension follows the descending order of KAc > FA > DMSO, implying that the hydrogen atoms of DMSO form stronger bonding with oxygen atoms of Si-OH in the inner surface of kaolinite. Further, they constrain the mobility of the oxygen atoms in the z dimension. To summarize, the MSD simulation results are in good accordance with the FTIR and RDF results.

#### 3.6.3. Elastic Properties

The elastic properties of the organo-kaolinite complexes calculated by MD simulation are listed in Table 3. The bulk modulus reflects the resistance of materials to uniform compression under the elastic system, which is mainly determined by the material chemical bond strength. Compressibility represents the compression properties of materials. The shear modulus indicates the shear resistance of the material [36,37].

It can be seen that the bulk modulus and the shear modulus of the three organo-kaolinite complexes decrease with the increase in the insertion number of organic molecules, while the compressibility increases with an increase in the intercalated molecules. The DMSO–kaolinite complex with the largest insertion rate has the largest compressibility, which is the smallest bulk modulus and shear modulus among the organo-kaolinite complexes. The KAc–kaolinite complex with the minimum number of inserted molecules exhibits the smallest compressibility, as well as the largest bulk modulus and shear modulus when compared with other complexes. The results suggest that in regard to the organo-kaolinite complexes with higher intercalated organic molecules, more hydrogen bonds between the kaolinite layers are broken. Under the action of shearing force, the kaolinite lamellas are easier to peel off and the shearing force resistance is relatively weak. On the other hand, with the increase in the organic molecule insertion amount, the interlayer space and the compressibility of the intercalated kaolinite increase, while the relative bulk modulus decreases. Kaolinite with a high content of intercalated molecules can weaken the surface structural damage caused by crushing force, thereby buffering the abrasion of the kaolinite crystal induced by ball milling and retaining the structural integrity of kaolinite.

## 4. Discussion

According to Figure 2, KAc was unable to be intercalated into the interlayer of kaolinite within a short time of wet ball milling. In contrast, DMSO and FA were easily inserted and thus were able to increase the interlayer spacing of kaolinite. For the molecules that were larger than FA, the interlayer spacing of kaolinite increased by DMSO was wider than that which were intercalated by FA. When combined with the analysis results of the property, the micro morphology and simulation of kaolinite with wet ball milling in different solvents, as well as the intercalation effect and mechanism of kaolinite in different solvents with wet ball milling, can both be graphically illustrated, as shown in Figure 10. The kaolinite in the solvent with no intercalation suffer more mechanical destruction. Further, the layered stacking structure of kaolinite is damaged and peeled off layer by layer. Numerous cracks and apparent damage arose on the surface of the flake structure; in addition, kaolinite were broken into fine fragments in the solvent. As for the solvent that can be intercalated in kaolinite by wet ball milling, it can buffer the mechanical damage by the intercalation of the organic molecules and thus maintain the integrity of the lamellar structure in order to obtain kaolinite nanosheets after wet ball milling. In the process of intercalation and exfoliation, the crystal structure of kaolinite with more solvent molecules intercalated into the interlayer is more complete than that with less insertion. The simulation results were consistent with the previous experimental results and could also prove the mechanism of intercalation.

## 5. Conclusions

The effects of organic intercalators on the structural integrity and properties of kaolinite by wet ball milling were investigated based on experimental and molecular dynamic (MD) studies. The wet ball milling process introduced shearing force and a squeezing destructive force in order to render the kaolinite sheet to be peeled off. During the intercalation and exfoliation process, the flakes of unintercalated kaolinite broke into small fragments that were attached on the kaolinite surface and interlayers, due to the destructive squeezing and shearing force induced by wet ball milling. When compared to KAc, the FA and DMSO molecules could be easily inserted into the interlayers of the acidified kaolinite under a wet ball milling process. The kaolinite intercalated with more organic molecules could buffer the crushing force by wet ball milling. In addition, the intercalated kaolinite achieved reduced thickness, increased specific surface area, as well as a relatively well integrated peeled layer when compared with the original kaolinite under a ball milling process. Based on the MD simulation results, the calculated elastic properties of the organo-kaolinite complex revealed that the interlayer space and the compressibility of the intercalated kaolinite increased, while the bulk modulus decreased with an increasing number of the guest organic molecules. The DMSO–kaolinite complex was easier to peel off compared to the other organic intercalators due to its more intercalated molecules.

## Figures and Tables

**Figure 1 nanomaterials-12-04255-f001:**
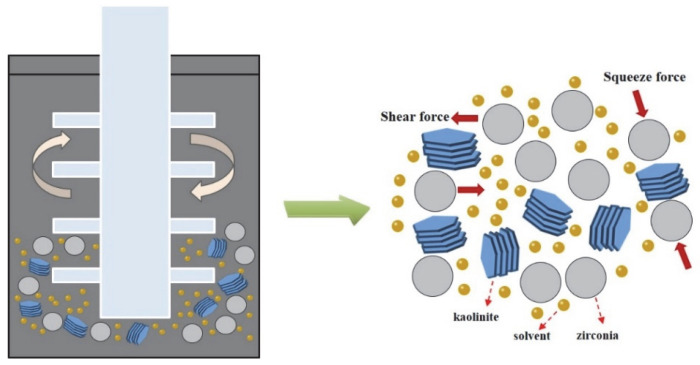
Intercalation process via wet ball milling.

**Figure 2 nanomaterials-12-04255-f002:**
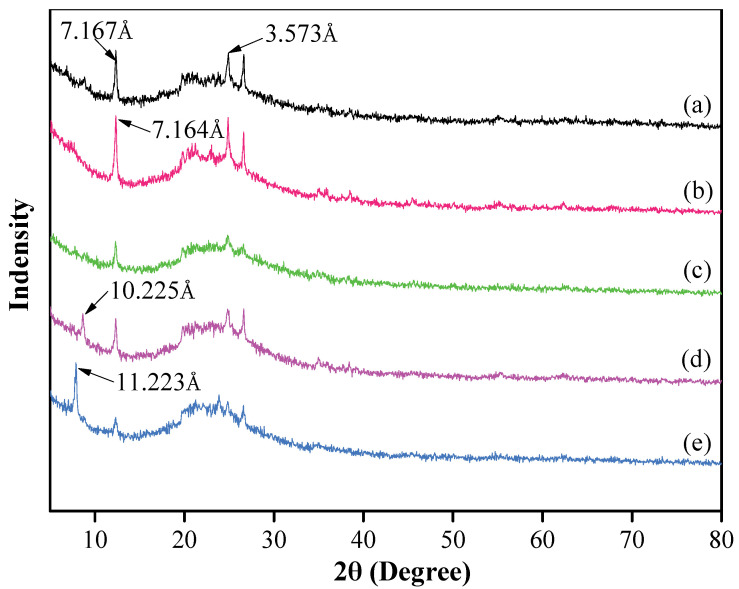
XRD patterns of (**a**) original kaolinite; (**b**) acidified kaolinite; (**c**) the Kac–kaolinite complex; (**d**) the FA–kaolinite complex and (**e**) the DMSO–kaolinite complex.

**Figure 3 nanomaterials-12-04255-f003:**
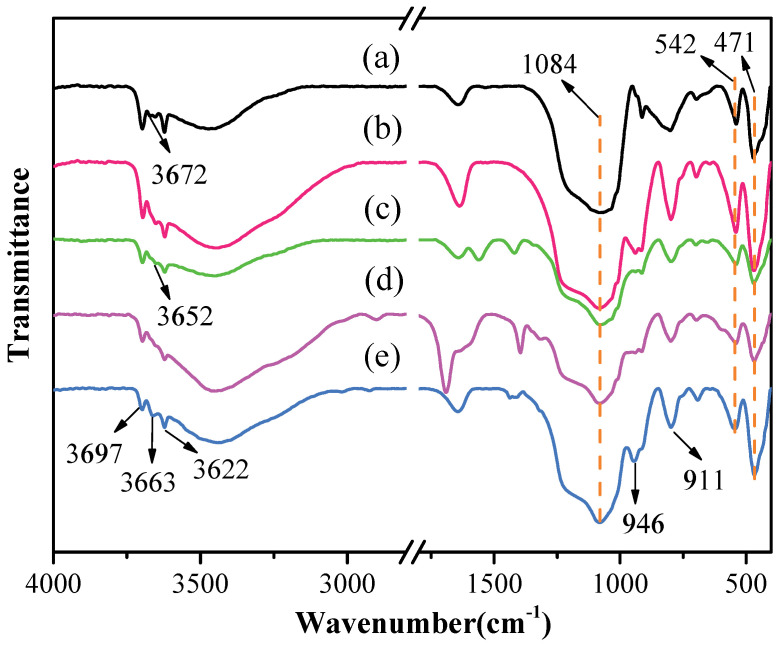
The FTIR spectra (400−4000 cm^−1^) of (**a**) original kaolinite; (**b**) acidified kaolinite; (**c**) the KAc–kaolinite complex; (**d**) the FA–kaolinite complex and (**e**) the DMSO–kaolinite complex.

**Figure 4 nanomaterials-12-04255-f004:**
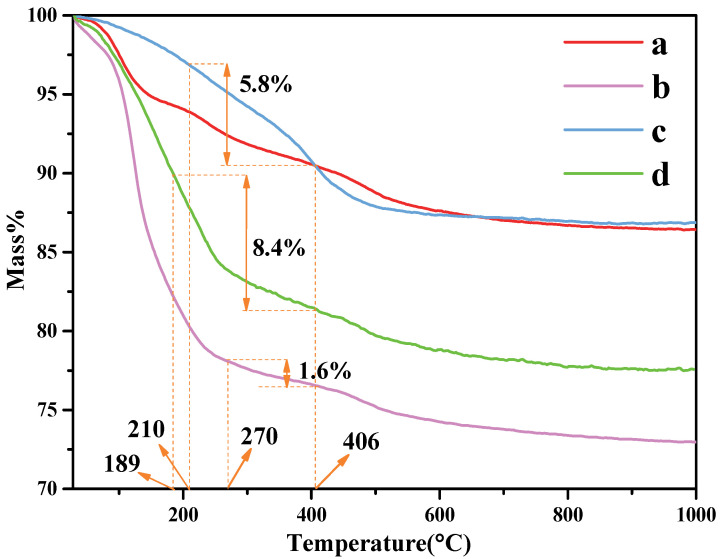
TG curves of (**a**) original kaolinite; (**b**) the KAc–kaolinite complex; (**c**) the FA–kaolinite complex; (**d**) the DMSO–kaolinite complex.

**Figure 5 nanomaterials-12-04255-f005:**
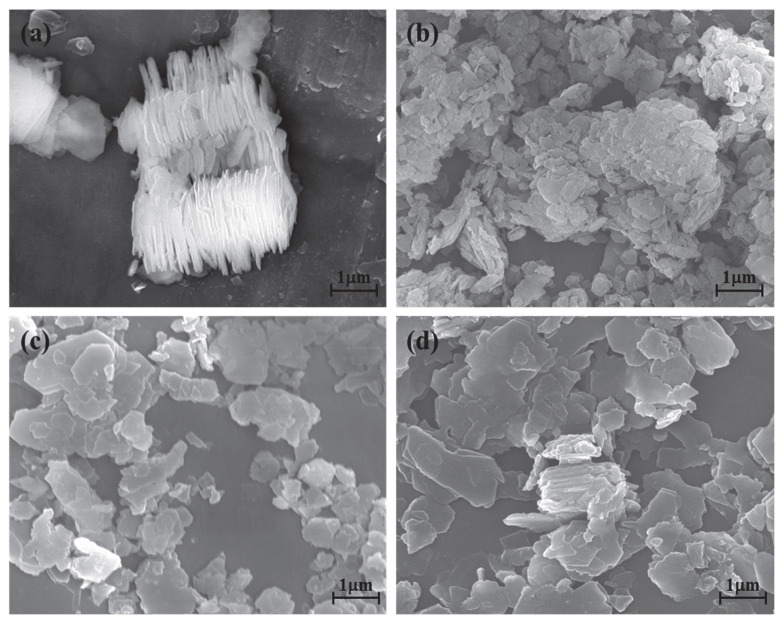
The SEM images of (**a**) original kaolinite; (**b**) the KAc–kaolinite complex; (**c**) the FA–kaolinite complex and (**d**) the DMSO–kaolinite complex.

**Figure 6 nanomaterials-12-04255-f006:**
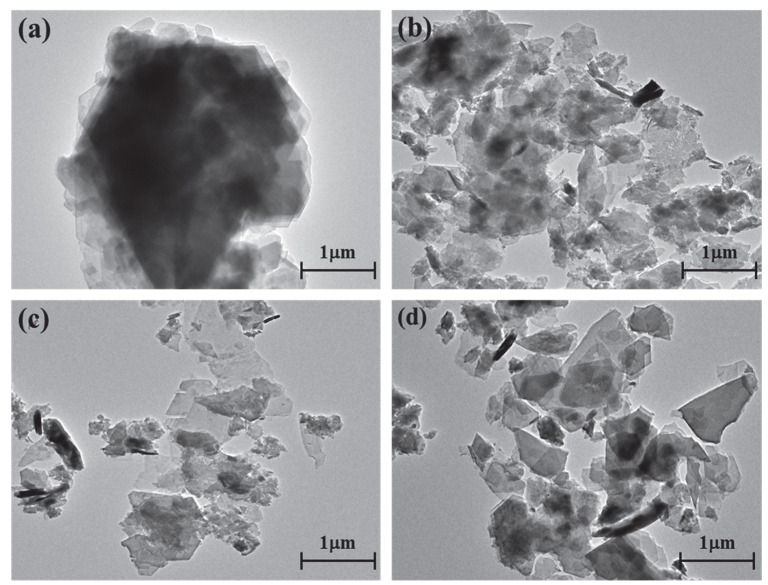
The TEM images of (**a**) original kaolinite; (**b**) the KAc–kaolinite complex; (**c**) the FA–kaolinite complex and (**d**) the DMSO–kaolinite complex.

**Figure 7 nanomaterials-12-04255-f007:**
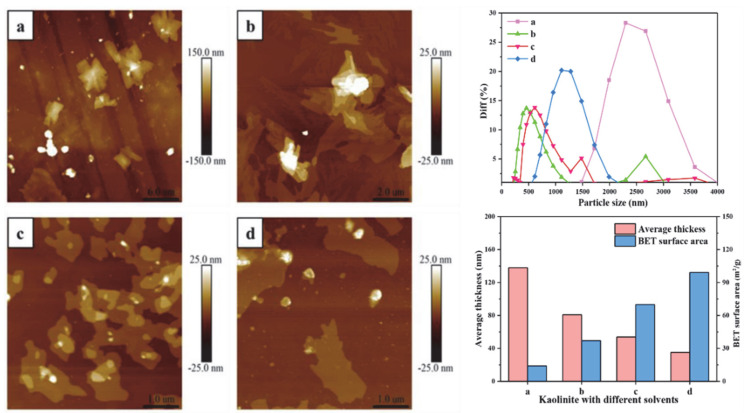
Nano particle size, thickness and BET surface area of (**a**) original kaolinite; (**b**) the KAc–kaolinite complex; (**c**) the FA−kaolinite complex and (**d**) the DMSO−kaolinite complex.

**Figure 8 nanomaterials-12-04255-f008:**
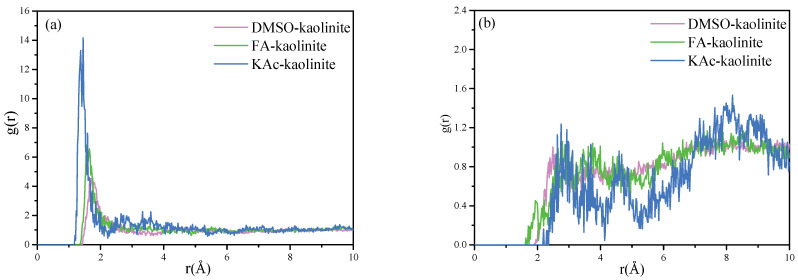
(**a**) RDF calculated for hydroxyl hydrogen atoms of the alumina octahedral surface of kaolinite to oxygen atoms of KAc, FA and DMSO and (**b**) RDF calculated for oxygen atoms of silicon tetrahedral surface of kaolinite to hydrogen atoms of KAc, FA and DMSO.

**Figure 9 nanomaterials-12-04255-f009:**
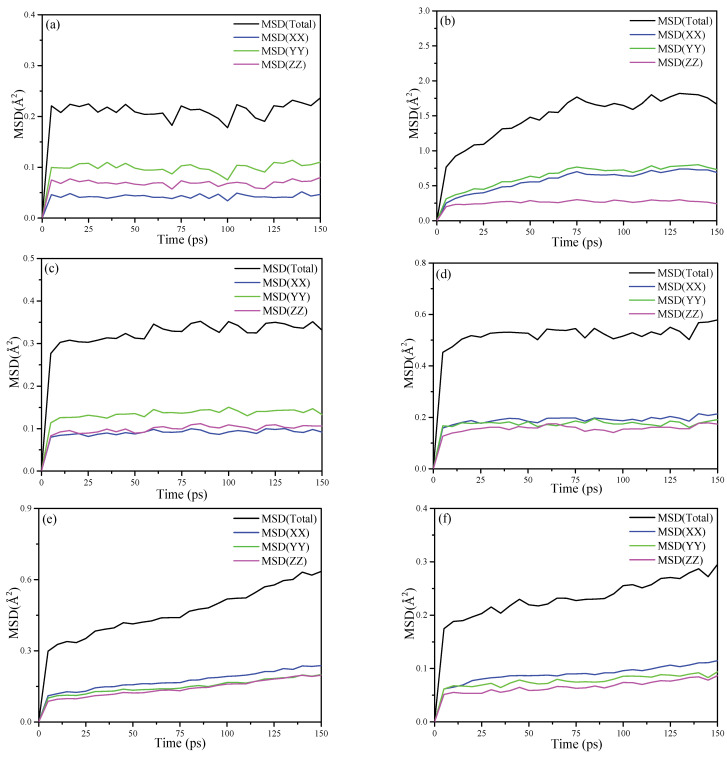
MSD calculated for (**a**) the oxygen atoms of KAc; (**b**) hydrogen atoms of KAc; (**c**) oxygen atoms of FA; (**d**) hydrogen atoms of FA; (**e**) oxygen atoms of DMSO and (**f**) the hydrogen atoms of DMSO.

**Figure 10 nanomaterials-12-04255-f010:**
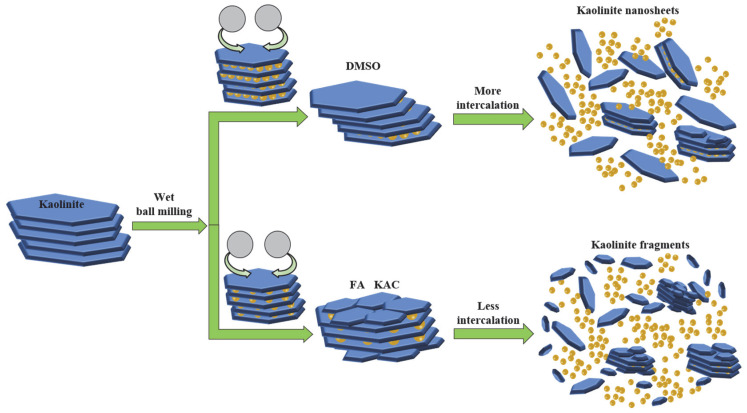
Intercalation effect and mechanism of kaolinite in different solvents with wet ball milling.

**Table 1 nanomaterials-12-04255-t001:** Chemical composition of kaolinite (wt.%).

SiO_2_	Al_2_O_3_	Fe_2_O_3_	K_2_O	TiO_2_	SO_3_	P_2_O_5_	CaO	CuO
52.19	42.92	0.73	1.95	0.37	0.24	0.14	0.93	0.02

**Table 2 nanomaterials-12-04255-t002:** Composition of the original kaolinite and the acidified kaolinite.

	SiO_2_	Al_2_O_3_	Fe_2_O_3_	K_2_O	TiO_2_	SO_3_	P_2_O_5_	CaO	CuO
Original kaolinite	52.19	42.92	0.73	1.95	0.37	0.24	0.14	0.93	0.02
Acidified kaolinite	60.72	34.85	0.28	0.32	0.49	0.11	0.06	0.05	0.01

**Table 3 nanomaterials-12-04255-t003:** The calculated elastic properties of the kaolinite intercalated complex.

Intercalated Complex	Bulk Modulus (GPa)	Shear Modulus (GPa)	Compressibility (1/TPa)
DMSO–kaolinite	30.6981	7.1807	32.5753
FA–kaolinite	34.7496	8.0201	28.7773
KAc–kaolinite	38.9959	8.0214	25.6437

## Data Availability

The data is included in the main text.

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
