# Peer review of "Effects of Typical Solvents on the Structural Integrity and Properties of Activated Kaolinite by Wet Ball Milling"

_nanomaterials, 2022, doi:10.3390/nano12234255_

Round 1
Reviewer 1 Report
The aim of this work is to functionalize a kaolinitewith intercalators by means of wet ball milling and to characterize the products through a number of experimental techniques such as XRD, XRF, FTIR, SEM, TEM, TGA... together with a computational simulation, i.e., molecular dynamics. There is a wealth of experimental approaches, so that the overall results should be considered satisfactory. Besides, the authors introduce an analysis of RDF computed via molecular dynamics, coupling the results with MSE analysis. In my opinion, the work lacks only a simple simulation of the FTIR spectrum, which would demonstrate the validity of the computations and -consequentely- the validity of the conclusions drawn from the analysis of MSD and RDF.
I have to point out just a little typo: pag 13, row 403: sovlent instead of solvent.
Author Response
Point 1: The aim of this work is to functionalize a kaolinite with intercalators by means of wet ball milling and to characterize the products through a number of experimental techniques such as XRD, XRF, FTIR, SEM, TEM, TGA... together with a computational simulation, i.e., molecular dynamics. There is a wealth of experimental approaches, so that the overall results should be considered satisfactory. Besides, the authors introduce an analysis of RDF computed via molecular dynamics, coupling the results with MSE analysis. In my opinion, the work lacks only a simple simulation of the FTIR spectrum, which would demonstrate the validity of the computations and -consequentely- the validity of the conclusions drawn from the analysis of MSD and RDF.
- Thanks for the comments. The validity of the computations and conclusions drawn from the analysis of MSD and RDF were based on the TGA, which could quantify the change of intercalation.
Point 2: I have to point out just a little typo: pag 13, row 403: sovlent instead of solvent
-Thanks for the comments. The authors apologize for the mistakes. We checked and modified these mistakes in the revised manuscript.

Reviewer 2 Report
I kindly ask authors to prepare a response letter point-by-point rebuttal and must be subjected to the manuscript as well, considering the following comments with sufficient explanations.
1) In all the text “Fig.“as abbreviation of “Figure” word has been used; in contrast, in the subtitle of each figure, the complete “Figure” word has been used. Therefore, they must be identical and everywhere through the text all “Fig.” must be changed to the “Figure”.
2) The introduction is very short and the properties of 2 different types of kaolinite and disordered kaolinite and their dehydroxylation (DHX) resulting productions of metakaolinite and metadiskaolinite at different temperatures through experimental and numerical investigations (DFT computational methods) must be reported, the following literature is used for the extraction of these information: DOI: 10.1007/s42860-020-00082-w.
3) I am wondering which type of Kaolinite (kaolinite or disorder in kaolinite) has been used for this study? It is highly suggested to use the XRD or NMR measurement for the samples, which have been used for this study to find out about the structure of stacked layers of MK. The reason is that the dissolution rate of Al is different for metakaolinte or metadiskaolinte or partially dehydroxylated metakaolinte and metadiskaolinte depending on the folded Al.
4) Cite the programs and properties, which have been used in the second paragraph from “2.4. Models and simulation details “. The Forcite program, Material studio software, und the properties ….!
5) The Properties of the XRD, FTIR, SEM, TGA instruments used for this study must be explained in the methodology to make it clear for readers. As an example for XRD and FTIR measurements, the following literature can help you. https://doi.org/10.1021/acs.jpcc.1c10151
6) "Authors contributions" after conclusion are not sated and must be included.
7) The fonts must be checked like lines 422 and 425.
8) T(ps) in horizontal axis of Figure 9 must be fully written “Time(ps)”.
Author Response
Response to Review 2 Comments
Point 1: In all the text “Fig.“as abbreviation of “Figure” word has been used; in contrast, in the subtitle of each figure, the complete “Figure” word has been used. Therefore, they must be identical and everywhere through the text all “Fig.” must be changed to the “Figure”.
-Thanks for the comments. All the “Figure” were changed to “Fig.” in the revised manuscript.
Point 2: The introduction is very short and the properties of 2 different types of kaolinite and disordered kaolinite and their dehydroxylation (DHX) resulting productions of metakaolinite and metadiskaolinite at different temperatures through experimental and numerical investigations (DFT computational methods) must be reported, the following literature is used for the extraction of these information: DOI: 10.1007/s42860-020-00082-w.
-Thank you very much for the advice. We added and referred the paper mentioned above, which was used for this study. The kaolinite in the paper was divided into three types (The original kaolinite, kaolinite with more intercalation, kaolinite with less intercalation), this is the focus of this article. Therefore, the properties of 2 different types of kaolinite and disordered kaolinite and their dehydroxylation (DHX) resulting productions of metakaolinite and metadiskaolinite at different temperatures through experimental and numerical investigations (DFT computational methods) were discussed and reported in the follow-up work
Point 3: I am wondering which type of Kaolinite (kaolinite or disorder in kaolinite) has been used for this study? It is highly suggested to use the XRD or NMR measurement for the samples, which have been used for this study to find out about the structure of stacked layers of MK. The reason is that the dissolution rate of Al is different for metakaolinte or metadiskaolinte or partially dehydroxylated metakaolinte and metadiskaolinte depending on the folded Al.
-Thanks for the comments. The XRD measurement was used for the samples as shown in Fig.2. The effects of typical solvents on the structural integrity and properties of kaolinite were discussed in this paper. The kaolinite used for intercalation was the same and the dissolution rate of Al was mainly affected by acidification.
Point 4: Cite the programs and properties, which have been used in the second paragraph from “2.4. Models and simulation details “. The Forcite program, Material studio software, and the properties ….!
-Thanks for the comments. We referred to the modeling thought and method from the cited literature. The forcite program, material studio software, and the properties were mentioned from line 152 to line 162 in the manuscript.
Point 5: The Properties of the XRD, FTIR, SEM, TGA instruments used for this study must be explained in the methodology to make it clear for readers. As an example for XRD and FTIR measurements, the following literature can help you. https://doi.org/10.1021/acs.jpcc.1c10151
-Thank you very much for the advice. We borrowed and cited the literature mentioned above. The properties of the XRD, FTIR, SEM, TGA instruments used for this study had been explained in the methodology to make it clear for readers. Some relevant sentences were added as follows: “The change of chemical properties of kaolinite before or after stripping was characterized by Fourier transform infrared spectroscopy (VECTOR-22IR, Bruker Instruments Corp). The powder samples of test were dried and mixed with KBr for tablet processing. The condi-tions of infrared test ranged from 400 cm-1 to 4000 cm-1, the test resolution was 4cm-1 and each test was scanned 64 times.”
Point 6: "Authors contributions" after conclusion are not sated and must be included.
-Thanks for the comments. “Authors contributions” after conclusion were added in the revised manuscript.
Point 7: The fonts must be checked like lines 422 and 425.
-Thanks for the comments. We checked and modified the fonts like lines 422 and 425 in the revised manuscript.
Point 8: T(ps) in horizontal axis of Figure 9 must be fully written “Time(ps)”.
-Thanks for the comments. T(ps) in horizontal axis of figure 9 was fully written “Time (ps)” in the revised manuscript.

Round 2
Reviewer 2 Report
Dear Authors,
Regarding Point 1, please check the template of Materials journal, I think the abbreviation must not be used as I explained in the first review report.
Regarding point 4, I don't see that you added any citations to the manuscript (softwares and methods must be cited), so please add some citations from the methods that you used for this study.
Then it will be accepted for publication.
Best regards,
Author Response
Response to Review 2 Comments
Point 1:Regarding Point 1, please check the template of Materials journal, I think the abbreviation must not be used as I explained in the first review report.
-Thanks for the comments. We checked the template of journal and modified the abbreviation in the revised manuscript.
Point 2:Regarding point 4, I don't see that you added any citations to the manuscript (softwares and methods must be cited), so please add some citations from the methods that you used for this study.
-Thanks for the comments. Some citations from the methods that we used for this study were added in the revised manuscript.
